# A Novel Approach to Reduce Sedentary Behaviour in Care Home Residents: The GET READY Study Utilising Service-Learning and Co-Creation

**DOI:** 10.3390/ijerph16030418

**Published:** 2019-02-01

**Authors:** Maria Giné-Garriga, Marlene Sandlund, Philippa M. Dall, Sebastien F. M. Chastin, Susana Pérez, Dawn A. Skelton

**Affiliations:** 1School of Health and Life Sciences, Glasgow Caledonian University, Cowcaddens Road, Glasgow G4 0BA, UK; Philippa.Dall@gcu.ac.uk (P.M.D.); Sebastien.Chastin@gcu.ac.uk (S.F.M.C.); Dawn.Skelton@gcu.ac.uk (D.A.S.); 2Department of Physical Activity and Sport Sciences, Faculty of Psychology, Education and Sport Sciences (FPCEE) Blanquerna, Ramon Llull University, Císter 34, 08022 Barcelona, Spain; SusanaPT@blanquerna.url.edu; 3Department of Community Medicine and Rehabilitation, Umeå University, 901 87 Umeå, Sweden; marlene.sandlund@umu.se; 4Department of Movement and Sport Science, Ghent University, St. Pietersnieuwstraat 33, 9000 Ghent, Belgium

**Keywords:** Co-creation, service-learning, care home residents, sedentary behaviour, physical activity

## Abstract

The GET READY study aimed to integrate service-learning methodology into University degrees by offering students individual service opportunities with residential care homes, to co-create the best suited intervention to reduce the sedentary behaviour (SB) of residents throughout the day, with researchers, end-users, care staff, family members and policymakers. Eight workshops with care home residents and four workshops with care staff, relatives and policymakers, led by undergraduate students, were audiotaped, transcribed verbatim and analysed with inductive thematic analysis to understand views and preferences for sustainable strategies to reduce SB and increase movement of residents. Perspectives about SB and movement in care homes highlighted four subthemes. Assets for decreasing SB included three subthemes, and suggestions and strategies encapsulated four subthemes. There is a need to include end-users in decision making, and involve care staff and relatives in enhancing strategies to reduce SB among residents if we want sustainable changes in behaviour. A change in the culture at a policymaker and care staff’s level could provide opportunities to open care homes to the community with regular activities outside the care home premises, and offer household chores and opportunities to give residents a role in maintaining their home environment.

## 1. Introduction

The number of older adults will increase significantly in the coming decades at a faster pace than any other age segment of the European Union’s population [1]. This increase is likely to be linked to a growing demand for long-term care, placing a significant strain on health care resources. One in four older adults will spend time in a care home and the need for such care will persist [2]. Care-home residents are amongst the frailest in our population with high levels of physical dependency [3], and with three-quarters having cognitive impairment [4]. This will require long-term care facilities’ policies to offer effective and sustainable interventions to address their complex physical and mental health needs [5].

Research over decades reports that care home residents spend the majority of their time inactive [6]. It is well known that regular physical activity (PA) limits the development and progression of chronic diseases and disabling conditions [7]. However, time spent in sedentary behaviour (SB) has increased substantially over the last three decades [8] and increases with age [9]. SB has been gaining recognition as a risk factor, sometimes independent to PA status, for numerous health conditions and reduced mobility [10]. Additionally, care home residents spend most of their waking day sedentary (e.g., sitting, watching television), with low levels of interaction with staff and each other [6]. Consequently, the National Institute for Health and Care Excellence [11] issued a quality standard call for “older adults in care homes to be offered opportunities during their day to participate in meaningful activities that promote their health and mental wellbeing”.

A typical day for a care home resident will be made up of a sequence of periods of SB, light-intensity PA (LIPA) and moderate-to-vigorous PA (MVPA) [12,13]. Care home residents spend on average 79% of their day sedentary, 20% in LIPA, and 1% in MVPA [14]. Understanding the optimal structure of time spent in various activities is desirable and might be of great importance when designing interventions to modify such behaviours. Recommendations could focus on reducing SB by introducing LIPA throughout the day, while this focus would contain two main messages: To sit less and move more [15]. There is still a gap in knowledge on how to change the mind set and activities offered to older residents by ‘gate-keeper’ health professionals and how to make ‘moving more often’ normal in the residential setting. Despite the growing interest in SB research, there has been a lack of studies focused on reducing SB and enhancing movement (as opposed to specific exercise interventions) in institutionalised older adults.

Patient and public involvement (PPI) is playing an increasingly important role in health and social care research and the design of service provision [16]. Despite the increasing emphasis on PPI, marginalised groups, such as care-home residents can be overlooked when including people in the research process, although a recent systematic review showed that care-home residents could be successfully involved [17]. Engagement is key and helps address the challenges related to translation and implementation in complex organisational settings [18]. Emergent from the participatory design paradigm is a process called co-creation [19], which is hypothesised to have a strong and enduring impact on health outcomes [20], and may be a promising strategy to address other complex health behaviours. This shifts the design process from the traditional “top-down” health model to an inductive paradigm of shared leadership allowing end-users to take control over the content of the activities [21], and be involved in their health management and decision-making relevant to their own health [22].

University degrees, such as Physical Therapy and Sport Sciences, need to have a practical approach rather than just be dominated by theory, translating what is learned in the safe and controlled classroom to what occurs in the wider working world. Service learning is a teaching and learning strategy that involves and integrates students in meaningful community service with academic instruction focusing on critical, reflective thinking to enrich the learning experience, teach civic responsibility, and strengthen communities [23,24]. To our knowledge, there is no previous experience involving university students and care home residents to co-create an intervention to reduce SB and enhance movement throughout the day. 

The GET READY study integrated a service-learning methodology into Physical Therapy and Sport Sciences University degrees by offering students individual service opportunities with residential care homes [25]. They were tasked to co-create the best suited intervention to reduce the SB of residents and enhance movement throughout the day, together with researchers, end-users, care staff members, family members and policymakers, which enriched their learning with a distinctly multi-disciplinary and inter-sectorial nature. We set the following two general research questions: (a) How are SB and movement in care homes perceived and experienced among care home residents, staff members, relatives and policymakers?; and (b) how can we work together to decrease sedentary behaviour and increase movement in care home residents?

## 2. Materials and Methods

### 2.1. Participants

Students were recruited to participate in the GET READY study from modules from degree programs within the School of Health and Life Sciences (Glasgow) and Sport Sciences (Barcelona). Care home residents, staff and family members were recruited by a contact staff member on a voluntary basis from two care homes in each country (a total of four care homes participated in the study). Policymakers were recruited through the research team. A contact staff member from each home purposively recruited residents who could engage in discussion with the students, with no exclusion criteria on health. The ethics committees of the first author’s institutions approved this study (Glasgow Caledonian University [GCU], and the Faculty of Psychology, Education and Sport Sciences Blanquerna [FPCEE]), and all participants (or legal guardians if unable to consent) signed an informed consent prior to participation.

Overall, 26 students from GCU and the FPCEE, designed and conducted two workshops for care home residents (WS1 and WS2) and one workshop for care staff, family members and policymakers (WS3) within each care home. Thus, a total of eight workshops were conducted with care home residents and a total of four were conducted with care staff, family members and policymakers.

WS1 and WS2 in all four care homes included a total of 22 residents (59% females, mean age 83.2 (11.6) years) (Table 1). WS3 included 14 care staff (7 senior care assistants, 3 physical therapists, 2 geriatricians, 1 occupational therapist, and 1 nurse), 4 family members (1 brother, 1 sister and 2 daughters), and 4 policymakers.

### 2.2. Study Design

A co-design process guided by participatory action research (PAR) methodology was applied. Students developed two workshops to be conducted with end users (care home residents), and one workshop to be conducted with staff and family members, and policymakers. These workshops were aimed at designing personalised strategies to reduce SB and enhance movement throughout the day tailored to the needs of each resident, based and focused on the end-user movement patterns, perspective, and willingness to change.

The three workshops aimed at answering the following four narrower research questions (from the two general research questions): (a) understand what care home residents, staff members, relatives and policymakers think about SB and movement in care homes, (b) collect views among care home residents, staff members, relatives and policymakers about their reasons for decreasing SB and increasing movement, (c) understand how care home residents, staff members, relatives and policymakers can make an impact reducing SB and increasing movement in a care home setting, and (d) collect ideas among care home residents, staff members, relatives and policymakers of strategies to decrease SB and increase movement within a care home setting. More information about the workshops’ contents can be found in the study protocol [25].

### 2.3. Data Collection

Care staff at the care homes provided clinical and demographic information to allow description of the care home residents (Table 1).

The service-learning methodology was integrated within current modules in GCU and FPCEE. Content related to service-learning methodology and successful experiences using a service-learning methodology for undergraduate students, how to co-create successful interventions, and tips on how to conduct a discussion group and foster positive group dynamics, were added to each module’s material. In each country, one group of students was involved in the design of two workshops (WS1 and WS2) for care home residents, and another group of students designed a workshop for staff members, relatives and policy makers (WS3) to gather similar information. Students conducted the workshop together with a researcher. WS1 and WS2 lasted between 38 and 55 minutes each and WS3 lasted between 57 and 75 minutes. Each workshop was audiotaped with participants’ consent.

Between WS1 and WS2, the care home residents were asked to wear an ActivPAL^TM^ monitor (PAL Technologies, Glasgow, Scotland), a valid “gold standard” method to measure sedentary behaviour [26,27]. The monitor recorded the total time the residents spent sitting, upright time and walking, for nine days continuously. A weekly graphical representation of SB data was fed back to participants at the second workshop, to raise their awareness of their sedentary time and their SB patterns, to enable identification of the best-suited strategy to modify this behaviour as it might be that certain times of the day are more appropriate for intervention.

### 2.4. Transcription and Data Analyses

Transcription of twelve workshops (WS1, WS2 and WS3 in four care homes) was completed verbatim by one researcher (M.G.-G.), and another subsequently performed spot checks on 50% of transcripts to ensure accuracy (D.A.S.). The workshops conducted in Barcelona were originally in Catalan and Spanish, and translated into English (M.G.-G.). To allow for the revelation of a shared phenomenon from the data, an inductive thematic analysis was conducted, following the six steps described by Braun and Clarke [28]: (1) The reading and re-reading of transcripts to achieve familiarisation with the data; assumptions and ideas that surfaced were also noted by researchers (M.G.-G., M.S., D.A.S.), (2) One researcher (M.G.-G.) made initial codes noting interesting features of the data, including quotes perceived as significant, (3) initial codes were then organised into meaningful groups—themes [29] (M.G.-G., M.S.), (4) then two researchers reviewed, defined and named their themes (M.G.-G., M.S.), (5) researchers collectively reviewed codes and themes; redefining, renaming, and collated themes when necessary; and, (6) after discussion, three themes were decided upon that were deemed to best represent the participants’ perspective. These were then reported back to the residents, staff and family members within the care homes to ensure there was agreement with the content of the themes.

## 3. Results

From the 12 workshops (eight workshops with care home residents, four workshops with care staff, family members and stakeholders), three themes emerged that encapsulated the participants’ perspectives: (1) Knowledge of and attitudes towards the behaviours to be tackled; (2) are PA/SB worth the effort? Assets for decreasing SB and increasing PA; and, (3) taking action: Suggestions and strategies. Table 2 shows the themes and subthemes identified within the workshops.

### 3.1. Knowledge of and Attitudes Towards the Behaviours to Be Tackled

This overarching theme encompassed the participants’ perspectives about SB and movement in care homes. Four subthemes were found to group their overall perspectives: ‘PA relates to health and happiness’, ‘PA is for everybody’, ‘overprotection is a barrier’, and ‘inactivity is in the walls’.

PA related to feeling better and to several health benefits, such as improvement of heart problems and less pain. Better mobility was identified by most of the participants. PA was also related to being lucky and the chances and opportunities one had in life (e.g., previous history of PA). On the contrary, several health issues were reported as common barriers to being sedentary and inactive, including pain, fatigue, mobility problems, weakness, depressive symptoms, or fear of falling: *“Even though I try to move as much as possible, there are times I cannot move because I have a lot of pain in my legs (…), I am so angry, I can’t move”* (female, 83 years old, Barcelona). The increasing movement was related to health-economic benefits among policymakers: *“It has been shown that increasing PA levels have a direct impact on healthcare savings in European countries”* (female, Public Health Agency, Barcelona).

A positive opinion of being physically active was shared among the participants, stating that PA should be important for everybody of all ages. A resident shared: *“I’m proud of being able to exercise at an older age”* (male, 71 years old, Glasgow). Having reduced mobility, such as being in a wheelchair was not perceived to be a barrier to being physically active. Some negative opinions of being too sedentary were also shared among residents: *“[A] sedentary person is a person who, in the end, slowly becomes silly. (…) Foolish in the sense of an older adult who does not move and does nothing”* (female, 84 years old, Barcelona), and *“a person who does not move his legs, does not move his head”* (female, 79 years old, Glasgow).

Residents and stakeholders seemed aware that care staff and relatives tended to be too protective as they were worried about residents being hurt, and this overprotection tended to limit PA: *“The most important thing is the safety of the patient, the resident, and I think it’s very admirable that people should be as mobile as possible but within the limits to keep them safe (…)”* (female, sister of a resident, Glasgow); *“Safety is important, a fall can knock your confidence”* (female, policy officer in the Active Scotland division in Scottish Government, Glasgow). It is worth noting that health professionals’ comments seemed to have an effect on residents’ decisions: *“[T]he doctor told me that I would feel better with the wheelchair”* (female, 76 years old, Barcelona), *“the doctor told me to quit going to the sport facility, he said it wasn’t appropriate for me”* (female, 73 years old, Glasgow).

Most residents spoke of a non-active/highly sedentary daily routine, almost an ethos of not moving, the fabric of the home was to sit—inactivity is in the walls: *“[A]fter breakfast I always go to my room and rest until lunch time”* (female, 103 years old, Glasgow), *“before living in the care home, I used to do all house chores that kept me more mobile and less sedentary”* (female, 83 years old, Barcelona). Residents stated they relied on care staff for most activities of daily living although they felt able to do much more: *“[H]e says not to do it, because it’s dangerous (...). And now I’m so angry and disappointed because they do not let me do so much, and I can, I know I can!”* (female, 83 years old, Barcelona).

### 3.2. Are SB/PA Worth the Effort? Assets for Decreasing SB and Increasing PA

This overarching theme included the subthemes of ‘longing for autonomy’, ‘wanting an improved wellbeing’, and ‘influence of significant others’.

The residents spoke of their wish for autonomy and being independent, willingness to be useful and to feel busy, as well as not wanting to bother anybody: *“I don’t want to see myself sitting in a wheelchair and being totally dependent on others”* (female, 83 years old, Barcelona). Negative feelings related to being too old, useless, hopeless, and not wanting to be in a care home, resulting in less movement, seemed common among residents: *“I think I have become to my friends a bit of a bore, I can’t help being angry most of the time for not being able to move”* (male, 99 years old, Glasgow), *“I just can’t believe that there isn’t anything I can do, I’m useless”* (female, 84 years old, Barcelona), *“I ended up here, it was not my decision to be here”* (male, 83 years old, Glasgow).

Most participants pointed out health-related reasons to increase PA and reduce SB, such as weight loss, improved diet and feelings of wellbeing: *“[W]hen exercising, I find myself better”* (male, 92 years old, Barcelona), *“I feel very well when I’m not sitting all day”* (female, 105 years old, Glasgow). Residents identified the benefits of moving more and sitting less for overall health: *“I think it is just better mobility, (…) exercise is really good, it keeps you going”* (female, 79 years old, Glasgow).

The influence of significant others like care staff and family members seemed to be an asset for reinforcing behaviour change: *“If my doctor thinks PA is good for my health then I sure need to do it”* (female, 74 years old, Glasgow). Some residents sought the acceptance of their family members and care staff, so they became a relevant influence on their behaviours. Similarly, family members encouraged movement when they perceived improvements among their relatives: *“My mom is always ready to participate in activities (…). She feels much better when she is able to do things on her own so I encourage her to do so”* (female, daughter of a resident, Barcelona), *“my sister takes me for a walk”* (female, 83 years old, Barcelona), *“my wife wants me to exercise”* (male, 77 years old, Barcelona).

### 3.3. Taking Action: Suggestions and Strategies

This theme encapsulates the suggestions among residents and relevant stakeholders (staff members, family members, policymakers) of how to reduce SB and increase movement throughout the day. Four subthemes were identified: ‘Involve residents in household chores’, ‘use regular reminders’, ‘engage end-users, family and staff members’, and ‘open up to the neighbourhood’.

The residents felt they had the capacity to get involved with household chores and tasks within the care home: *“I would like to help out with different tasks, I want to feel useful, help out. (…) There are a lot of tasks that can be done, and some of us could help and that would keep us moving more often”* (female, 103 years old, Glasgow), as well as maintain some activities they used to do: *“I used to dance sardanes (typical Catalan dance) and here I can’t do it and I would like to”* (female, 87 years old, Barcelona). Some staff members, however, pointed out some concerns about residents getting involved in household chores: *“A lot of safety and hygiene regulations in long-term care facilities have been applied in the past years that make it difficult for residents to engage in certain household activities”* (female, physiotherapist, Barcelona). The increasing movement could be included within the daily routine: *“It can be as simple as washing your own face with a facecloth. (...) Simple things like doing sit to stands when there’s an advert on the TV. They can all be really great”* (female, Care Inspectorate, Glasgow).

Residents agreed in the importance of having frequent reminders and encouragement to move more: *“It helps me when I’m told to stand up and walk (…), they do not let me sit for a long time”* (female, 87 years old, Barcelona), *“when it is nice out, they tell us to stand up and go for a walk in the garden, I like that”* (female, 79 years old, Glasgow). Staff and family members could identify ways to increase movement within daily routines, with regular reminders, as well as suggest activities that residents like to do and are capable of doing.

Most residents said they were happy to be involved in the design of strategies to move more and sit less. Some pointed out the importance of the strategies aimed at enhancing movement to have a social component and to be safe: *“I like doing activities with other people”* (female, 79 years old, Glasgow), “*I like to walk with a friend, my son doesn’t want me to go alone”* (female, 82 years old, Barcelona).

Regarding the ‘open up to the neighbourhood’ subtheme, a wide variety of activities were raised by the care home residents most of which could be added to their daily routines, such as walking to a particular place with a reason (e.g., going shopping to the supermarket), activities outside the care home to be aware of what is happening in the world and perhaps activities that enhance cognitive performance: *“He likes to get out and see what’s happening, get some fresh air”* (Female, sister of a resident, Glasgow), *“once a week we walk to a nearby park and do some exercise”* (female, 82 years old, Barcelona), *“I love it when we go out to get some fish and chips, I used to do that all the time before”* (male, 94 years old, Glasgow). Some stakeholders from both countries raised awareness of the closed environment of the care homes: *“[T]he care home structure should be rethought (…), we tend to function independently of what’s happening out there, so that our residents become more and more isolated*” (female, physical therapist, Barcelona), *“Care homes should be opened to the neighbourhood and closest facilities”* (male, senior care assistant, Glasgow). Some participants pointed out the importance of having an adapted and safe environment to walk indoors and outdoors.

After analysing the workshops and based on the literature we have suggested some strategies that could facilitate the co-design of a resident-centred intervention to reduce sedentary time and increase movement throughout the day. Thus, a summary of the challenges we aimed to face within this study, relevant quotes, and strategies proposed by the research team are shown in Table 3.

## 4. Discussion

The purpose of this study was to co-create, together with care home residents, students, care staff members, family members and policy makers, the best suited intervention to reduce the SB of care home residents and enhance movement throughout the day. To the knowledge of the authors, this is the first study to involve undergraduate students in a co-creation process with care home residents. The perspective of all stakeholders provided valuable insights for informing future sustainable strategies, as well as leading to a change in the culture of professionals working with this frail and co-morbid population. We also aimed to raise awareness, knowledge, skills and passion of graduates entering the workforce.

Existing research regarding physical rehabilitation for older people in long-term care has primarily involved the delivery of time-limited interventions (e.g., exercise classes) with limited attendance [30]. It seems necessary that, if such interventions are to be successfully and sustainably delivered, they need to be embedded in routine practice and that care home staff and relatives should be involved in developing and delivering the necessary change [31]. Further, there is increasing emphasis on programmes which reduce the overall time spent sedentary [32] and that do not simply involve short bursts of formally organised PA, such as exercise classes. Together, these reinforce the need for action to increase levels of PA in care homes, reduce time spent sedentary, and incorporate greater engagement of care home staff and relatives in developing and delivering whole practice change, which should be embedded in daily life routines.

Recent recommendations for PA in older adults state that reducing SB could be achieved by introducing light activity throughout the day [15]. This focus would contain two messages: To sit less and move more. Care staff could agree with each resident how this might happen, using motivational strategies, such as goal setting and self-monitoring [33]. Person-centred care, particularly fulfilling personal care and recreation preferences, and social-affective needs of long-term care residents could be applied [34]. Advice on how to accumulate time spent in light activity could include getting up from the chair and moving during television commercial breaks, adding some household chores, and encouraging five-minute walks throughout the day with family members or peers. Some non-active activities were reported as preferred for several residents (e.g., reading a book, watching television, and playing board games, such as cards or dominoes), and there was some discussion about prompting movement at appropriate points (end of a game, end of a chapter of a book).

Several health-related issues, such as pain, fatigue, mobility problems, weakness, depressive symptoms, or fear of falling were reported as barriers to doing PA and reducing SB. Older adults most often reported poor health as their primary barrier to move more, along with to fear of falling or injury, symptoms of depression, and a general disinterest in being active [35]. Reduced mobility, pain, and other symptoms of medical problems can affect an older adult’s ability and/or motivation to engage in PA [36]. It is worth noting that 63.6% of the residents (14/22) used a wheelchair during the workshops, although some could walk with assistance. Physical disability can be caused by (and result in) pain, which is experienced by 45% to 80% of care home residents [37]. However, being more physically active and less sedentary was related for most of our participants to several health benefits and feeling better overall. Evidence suggests that older adults are aware of the health benefits of PA, so much so that improving health was the most commonly reported reason older adults gave for engaging in PA [36,38,39]. Care staff and relatives could use this duality to their advantage by encouraging residents who report physical health as a barrier to PA by reminding them of its overall health benefits. When fatigue is the major complaint, suggestions include scheduling PA in short bouts [40].

Negative feelings related to being useless, helpless or unwilling to be at the care home seemed common among residents. Studies have regarded the move from a person’s own home into residential care as potentially traumatic, where residents are at risk of leaving aside their everyday routines and losing their identity [41]. Residents reported frustration around their lack of influence and independence in previous studies [42]. Everyday practices, such as shopping, cooking and cleaning, had been identified in the present study as being important to feel useful and maintain their daily activities, reported as well in previous studies [43]. Findings also support that residents of long-term care facilities who engage continuously in meaningful activities adjust better psychologically and socially to their new life in these facilities [44]. Residents in our study also reported paternalistic communication styles among staff, which tended to overtake daily tasks that could be done for themselves, again reported in a previous study [45]. As continuity of participation in meaningful activities is important for successful aging, care staff could easily encourage and support such activities within the everyday routines of the care home.

Residents discussed not only a loss of previous household routines (that prompted movement), but also a loss of connection with the wider community. Care staff pointed out the isolation residents tend to feel when living in a care home, as care homes tend to be unconnected with the rest of the world and feel far away from the community. Townsend [46] conducted an extensive survey of residential institutions and homes for the older adult population in England and Wales. He described a variety of negative effects associated with institutional relocation, including loss of occupation, isolation from family, friends and community, the tenuousness of new relationships, loneliness, loss of privacy and identity, and the collapse of self-determination. These same concerns are still prevalent today, and in some ways have become magnified within care home settings, with the increasing frailty and chronic health conditions of residents [47]. A change in the culture at a policymaker and care staff level could provide opportunities to open care homes to connect to the community with regular activities offered to residents outside the care home premises and events that bring the local community into the care home.

Feelings of loneliness leading to feelings of depression were common among residents in the present study (depression and/or anxiety was reported in 40.9% of the residents). The loss of care home residents’ self-determination due to institutionalisation was strongly related to loneliness and grief in a recent study [48]. Failure to find meaningful connections shows growing concerns about the critical rates of loneliness in residential care [49]. Many residents have trouble making meaningful social connections without support [50] and those living with dementia may have additional challenges due to increased difficulties in communication [51]. In a systematic review of qualitative studies on living well in various types of care homes, connectedness with others and caring practices emerged as two out of four key themes [52]. This indicates that care home staff members should encourage all kinds of initiatives that may strengthen residents’ coping resources, and this should include communicating with residents to reduce their loneliness and confirm their identity.

Having students in charge of leading the workshops had both strengths and limitations. On the one hand, students had a unique opportunity to work with care home residents and relevant stakeholders to understand their opinions regarding PA and SB, and their preferences on how to reduce SB and enhance movement, with a real-life workforce experience. Most residents seemed confident and comfortable with a student as the interviewer. However, due to the students’ lack of expertise conducting focus groups, some of the discussions were in less depth than they could have been, as in some cases they didn’t continue a topic when there was an interesting reply from a resident or other stakeholder. Loss of concentration and tiredness of some care home residents reduced the duration of workshops, thus some topics could not be covered.

Methodological discussions are both theoretical and practical in nature. We faced several challenges in the present study, such as: Involving undergraduate students in the co-creation process (being in charge of designing and conducting the workshops as a workforce experience through service-learning methodology); involving care home residents in the co-creation process (including those with mild to moderate dementia); involving busy care staff and relatives; facing suggestions of lack of communication (or miscommunication) between residents and care staff members; facing residents’ lack of confidence and feelings of being useless; trying to find a feasible and sustainable way to offer strategies that care home residents like to do and are able to do; trying to consider increased isolation and the opportunities and challenges that opening up a care home to neighbourhood resources might present.

## 5. Conclusions

The strategies raised by care home residents, care staff, relatives and policymakers are noteworthy and can serve as a guide for the design of a resident-centred intervention to reduce SB and enhance movement of care home residents. Undergraduate students can be successfully involved in a co-creation process within a care home setting to raise their awareness, knowledge, skills and passion before entering the workforce, within a sustainable framework.

These data demonstrate the importance of collaboration between end-users, care staff and relatives working together to find the best individualised approach to decrease SB and increase movement throughout the day. Care homes residents’ most common fear was to be dependent on others, and to be isolated in a closed environment. A change in the culture at a policymaker and care staff’s level could provide opportunities to open care homes to connect to the community with regular activities both inside and outside the care home premises, and offer household chores opportunities to residents to involve them in the everyday routine of their care home, as they would have done when independent living.

## Figures and Tables

**Table 1 ijerph-16-00418-t001:** Characteristics of care home residents (*n* = 22).

Characteristic	Glasgow (*n* = 12)	Barcelona (*n* = 10)	Total (*n* = 22)
Women, *n* (%)	6 (50)	7 (70)	13 (59)
Age, mean (SD)	83 (14.1)	83.4 (9.4)	83.2 (11.6)
Age, range	71–105	72–100	71–105
Marital status, *n* (%)			
Single	2 (16.7)	1 (10)	3 (13.6)
Married	3 (25)	4 (40)	7 (31.8)
Widow	6 (50)	5 (50)	11 (50)
Divorced	1 (8.3)	0	1 (4.6)
Diagnosis, *n* (%)			
High blood pressure	5 (41.7)	4 (40)	10 (45.5)
Stroke	1 (8.3)	1 (10)	2 (9.1)
Arthritis (rheumatoid and osteoarthritis)	0	4 (40)	4 (18.2)
Chronic obstructive pulmonary disease	1 (8.3)	1 (10)	2 (9.1)
Congestive heart failure (or heart disease)	1 (8.3)	1 (10)	2 (9.1)
Diabetes types I and II	3 (25)	4 (40)	7 (31.8)
Visual impairment (such as cataracts, glaucoma, macular degeneration)	2 (16.7)	5 (50)	7 (31.8)
Hearing impairment (very hard of hearing, even with hearing aids)	3 (25)	1 (10)	4 (18.2)
Degenerative disc disease (back disease, spinal stenosis, or severe chronic low back pain)	2 (16.7)	1 (10)	3 (13.6)
Depression	2 (16.7)	4 (40)	6 (27.3)
Anxiety	2 (16.7)	1 (10)	3 (13.6)
Cancer	3 (25)	2 (20)	5 (22.7)
Dementia or related illness	1 (8.3)	7 (70)	8 (36.4)
Number of current medications, mean (range)	8.5 (3–14)	10.4 (6–15)	9.3 (3–15)
SBQ, mean (SD)			
Hours sitting on a week day	8 (2.8)	7 (4)	7.6 (3.2)
Hours sitting on a weekend day	8.1 (3)	7.1 (2.8)	7.7 (2.9)
IPAQ, mean (SD)			
MET min/week	840 (713.1)	519.2 (606.1)	739.8 (678.6)

SD: Standard Deviation; SBQ: Sedentary Behaviour Questionnaire; IPAQ: International Physical Activity Questionnaire; MET: Metabolic Equivalent of Task.

**Table 2 ijerph-16-00418-t002:** Themes and subthemes of the 12 workshops.

Themes	Subthemes
1. Knowledge of and attitudes towards the behaviours to be tackled	1.1. PA relates to health and happiness. 1.2. PA is for everybody. 1.3. Overprotection is a barrier. 1.4. Inactivity is in the walls.
2. Are PA/SB worth the effort? Assets for decreasing SB and increasing PA	2.1. Longing for autonomy. 2.2. Wanting an improved wellbeing. 2.3. Influence of significant others.
3. Taking action: Suggestions and strategies	3.1. Involve residents in household chores. 3.2. Use regular reminders. 3.3. Engage end-users, family and staff members. 3.4. Open up to the neighbourhood.

**Table 3 ijerph-16-00418-t003:** Summary of the challenges to be faced, quotes and strategies proposed.

Challenge	Quotes	Strategy
Involving students in the co-creation process	*“(…) students tend to take more risks, they are younger and less worried (…), they tend to push our residents harder”* (female, occupational therapist, Barcelona). *“I like having young kids around, it feels nice to have new faces around, I feel comfortable”* (male, 83 years old, Glasgow).	Integrate undergraduate students in co-creating interventions with care home residents.
Involving end-users (care home residents) in the co-creation process	*“I like when someone asks me what do I want to do, I’m a grown-up and I want to decide, (…) I want to be taken into account”* (female, 84 years old, Barcelona).	Include end-users in decision making regarding their health. Make end-users responsible for their own health.Within a face-to-face interview decides between 2 and 4 strategies to reduce sedentary time and enhance movement with a care home staff member and a relevant family member. Use flipcharts and place them in the resident’s room so that they are constantly aware. Review once every two weeks, and modify accordingly.
Involving care staff and relatives in the co-creation process	Care staff views are important for most care home residents: *“If my doctor thinks PA is good for my health then I sure need to do it”* (female, 74 years old, Glasgow). Family members tend to encourage movement and seem to be an important influence on the residents’ behaviour change: *“[M]y wife bought me a pair of pedals to cycle while watching television”* (male, 77 years old, Barcelona), *“my daughter does not let me watch television all afternoon, she makes me stand up”* (male, 94 years old, Glasgow).	Involve care staff and relevant relatives to enhance strategies to reduce SB and increase movement in residents.
Communication between residents and care staff members is not always fluent	We detected a disconnect of points of view between some care home residents wishes and care staff actions in accordance. *“I feel I am never active now, …. So if anybody came up with something I would like to do, I would be delighted”* (male, 94 years old, Glasgow). *“We offer a lot of active activities (…), but it feels most residents are not interested”* (female, senior care assistant, Glasgow).	Within a face-to-face interview decides between 2 and 4 strategies to reduce sedentary time and enhance movement with a care home staff member and a relevant family member. Use flipcharts and place them in the resident’s room so that they are constantly aware. Review once every two weeks, and modify accordingly.
Not all care homes offer regular physical activity opportunities	If interventions to increase movement and reduce sedentary time are to be successfully and sustainably delivered, they need to be embedded in routine practice: *“(...) there is a place for, like, seated exercise classes but also there’s just this bigger place for the day to day - walking to the toilet and just being as active as possible in normal daily tasks”* (Female, policy officer in the Active Scotland division in Scottish Government).	Incorporate greater engagement of care home staff and relatives in developing and delivering whole practice change which should be embedded in daily life routines.
Enhance care home residents’ feelings of being useful	*“I don’t want the help from anyone. I rather do it myself if I can”* (female, 84 years old, Barcelona). *“If I could, I would go shopping for groceries and cook every day”* (female, 76 years old, Barcelona). *“I would like to help out with different tasks, I want to feel useful, help out. (…) There are a lot of tasks that can be done, and some of us could help and that would keep us moving more often”* (female, 103 years old, Glasgow). *“We have a thing called ‘My Home Life’ where it’s geared towards promoting their independence, which I totally believe in”* (male, care assistant, Glasgow).	Offer ‘household chore’ opportunities to residents to involve them in the everyday routine of the care home and allow them to feel useful and maintain some of their previous roles.
Find activities care home residents like to do/want to do	*“I like taking care of the garden”* (male, 79 years old, Barcelona). *“I was a great sailor and I love to row”* (male, 92 years old, Glasgow). *“I want to improve my memory, (…) I feel I’m losing it as days go by”* (female, 74 years old, Glasgow). *“the first thing is the newspaper, I like keeping updated with the world”* (female, 94 years old, Barcelona).	Within a face-to-face interview decides between 2 and 4 strategies to reduce sedentary time and enhance movement with a care home staff member and a relevant family member. Use flipcharts and place them in the resident’s room so that they are constantly aware. Review once every two weeks, and modify accordingly.
Open care homes to the neighbourhood	*“the care home structure should be rethought (…), we tend to function independently of what’s happening out there, so that our residents become more and more isolated”* (female, physical therapist, Barcelona). *“Care homes should be opened to the neighbourhood and closest facilities”* (male, senior care assistant, Glasgow).	Activities and synergies with nearby local facilities should be explored and offered to care home residents to battle isolation. A wide variety of activities were raised by the care home residents most of which could be added to their daily routines, such as walking to a particular place with a reason (e.g. going shopping to the supermarket), activities outside the care home to be aware of what is happening in the world, and establishing a routine to move.

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
