# Peer review of "A Novel Approach to Reduce Sedentary Behaviour in Care Home Residents: The GET READY Study Utilising Service-Learning and Co-Creation"

_ijerph, 2019, doi:10.3390/ijerph16030418_

Round 1

Reviewer 1 Report

Well written article.  

I was confused by the number of workshop interventions which were completed for the study.  The abstract says "Twelve workshops with home care residents, and four workshops with care staff, relatives, and policymakers."  The first line of the results section says "From the 12 workshops (8 workshops with home care residents, 4 workshops with care staff, family members and stakeholders)..."

This needs to be clarified - were there 16 workshops conducted (12 + 4) or 12 total workshops (8 + 4)???

I feel this needs to be more carefully explained, as it is confusing within section 2.1 and 2.2.  Perhaps more logically spelling out:  Two care homes were studied in both Glascow and Barcelona, for a total of four total care homes.  Workshops were done with two sets of care residents in each  of the four facilities, for a total of 8 workshops with residents.  A different workshop was designed for family members and stakeholders.  This workshop was done once at each of the four homes for a total of four workshops with these groups.

Other than that, this article was well written and included note-worthy information.

Author Response

We appreciate your comments and we would like to thank you for the time taken to review our work. Please find below a point-by-point response to your comments. You can find the changes in the manuscript underlined in yellow.

-        I agree that the explanation regarding the amount of workshops was confusing, and we have tried to clarify this point in the abstract and in the methods section, accordingly:

Abstract (lines 22-24):

“Eight workshops with care home residents and four workshops with care staff, relatives and policymakers, led by undergraduate students, (...)”

Methods section (lines 100-102): “Care home residents, staff and family members were recruited by a contact staff member on a voluntary basis from two care homes in each country (a total of four care homes participated in the study)”

Methods section (lines 110-111): “Thus, a total of eight workshops were conducted with care home residents and a total of four were conducted with care staff, family members and policymakers.”

Methods section (line 153): “Transcription of twelve workshops (...)”

Methods section (lines 169-170): “From the 12 workshops (8 workshops with care home residents, 4 workshops with care staff, family members and stakeholders)”

Reviewer 2 Report

row 100 onwards: "recruited by a..." how was prevented a selection bias?  similarly on row 102 what were the recruitment criteria for policimaker and all partecipants? Sampling is not explicitated: what is the percentage of students, senior care assistants, .... involved with respect to their total?. 

in table 1 columns with the characteristics of the overall home care residents should be added.

In the discussion a reference shuld be made on how many residents are able to walk autonomuosly and to the level of home residents disability (for istance a Barthel or a FIM could be provided to quantify it).

No mention was made of the effect on student training.

row 387 "... demonstrate importance of collaboration..." is poorly  supported by the presentation of the results: it is necessary to highlight the affirmations that support it and its quantity 

Author Response

We appreciate your comments and we would like to thank you for the time taken to review our work. Please find below a point-by-point response to your comments. You can find the changes in the manuscript underlined in yellow.

-       The authors agree with the reviewer’s comment regarding recruitment. Care home residents, staff and family members were recruited by a contact staff member on a voluntary basis. A selection bias is a limitation of the present study; however, a randomized selection is very unlike to occur in this setting and with the present study’s methodological approach. We asked the care staff member in charge of recruiting the residents to make sure residents were selected with different functional and activity levels, as well as with different levels of motivation towards activities’ engagement. The only inclusion criteria was to being able to follow the workshop, so that residents with severe dementia were excluded.

-       Policymakers were selected by the research team and we were willing to invite acknowledge professionals in the field of Health Promotion, Social Care and Public Health, and with a health management background able to give us a different insight on how to manage interventions to enhance movement and decrease sedentary behaviour in a care home setting.

-       The percentage of students that participated in the workshops was 100%. We conducted the service-learning methodology in one module within the School of Health and Life Sciences in Glasgow and one module in the Sport Sciences degree from Blanquerna (Barcelona), with a small number of students. This study was one of the tasks they were asked to do within the module. In the future, we aim to offer this structure in other modules with more students involved, and with more care homes joining the effort.

-       Table 1 shows the characteristics of the care home residents. We added three columns, the first one with the residents from the two care homes in Glasgow, the second column with the residents from the two care homes in Barcelona, and the third one with the overall sample from the four care homes. I’m afraid I don’t understand what the reviewer means by the overall care home residents? I believe this information is shown in our third column.

-       As suggested, we have added a sentence about the percentage of residents that needed a wheelchair (lines 319-320). Unfortunately, we didn’t collect information regarding their levels of disability, which might be worth adding in future studies. This later information is collected in different ways across care homes.

-       We totally agree that gathering the inputs of some students about their experience would have been very interesting. We are aiming to collect this information in the near future.

-       The authors appreciate your latest comment regarding the support of one of our conclusions, and we have referred it to table 3, where some quotes can be found (e.g. Involving students in the co-creation process, involving end-users (care home residents) in the co-creation process, and involving care staff and relatives in the co-creation process). A co-author of the present manuscript who is an expert in qualitative analysis (M.S.), emphasized not to provide a quantitative focus of the results, rather than the importance of each theme/subtheme.

We hope you will find our responses suitable. Please don’t hesitate to contact me for further clarification.